Relationship between the distribution of vegetation and the environment in the coastal embryo dunes of Jalisco, México

http://orcid.org/0000-0002-0178-4308 Frías-Ureña Héctor Gerardo 1
http://orcid.org/0000-0002-7945-8107 Ruiz-Corral José Ariel 1 ariel.ruiz@academicos.udg.mx
http://orcid.org/0000-0002-0609-615X Macías-Rodríguez Miguel Ángel 1
Durán Noé 2
Gonzalez Diego 2
http://orcid.org/0000-0001-9981-4757 De Albuquerque Fabio 3
Torres Morán José Pablo 2
1 Departamento de Ciencias Ambientales, Universidad de Guadalajara , Zapopan, Jalisco , México
2 Departamento de Producción Sustentable, Universidad de Guadalajara , Zapopan, Jalisco , México
3 School of Life Sciences Interdisciplinary Graduate Faculty, Arizona State University , Phoenix, Arizona , United States
Li Chenxi
Electronic publication date: 2022 Mar 2
Publication date: 2022
Volume: 10
Electronic Location ID: e13015
Received 2021 Jul 25; Accepted 2022 Feb 6
Copyright: © 2022 Frías-Ureña et al.
Copyright year: 2022
Copyright holder: Frías-Ureña et al.
License: This is an open access article distributed under the terms of the Creative Commons Attribution License, which permits unrestricted use, distribution, reproduction and adaptation in any medium and for any purpose provided that it is properly attributed. For attribution, the original author(s), title, publication source (PeerJ) and either DOI or URL of the article must be cited.
License URL: https://creativecommons.org/licenses/by/4.0/

Keywords: Soil formation, Mobile coastal dunes, Magnetite content, Dunes vegetation, Soil-plant relationships, Mexican Pacific

Funding: University of Guadalajara and Consejo Nacional de Ciencia y Tecnología, México This work was supported by the University of Guadalajara and Consejo Nacional de Ciencia y Tecnología, México. The funders had no role in study design, data collection and analysis, decision to publish, or preparation of the manuscript.

==============================
Background

The poorly developed soils of the embryo dunes imply little capacity for plant support, however, the adaptation mechanisms of plants respond sensitively to environmental variations, even when these variations are small, which results in a set of specialized habitats and flora that are rarely shared with other terrestrial ecosystems. The coastal dunes of the Mexican Pacific remain vaguely studied, this is why this research explored the relationship between environmental properties and the presence of plant species in the embryo dunes of the coast of Jalisco, Mexico.

Methods

Twenty-nine sites were sampled, one or two sites per embryo dune, with a random stratified design. Geomorphological and vegetation data were collected at site. Laboratory determinations included soil color, particle size, organic matter, pH, electrical conductivity, magnetite content, and moisture retention. Statistical analysis included correlation analysis to identify relationships between environmental variables; principal component analysis (PCA) and cluster analysis to group dune sites by environmental properties; canonical correspondence analysis (CCA) to determine a possible significant relationship between the presence of plant species and environmental variables; cluster analysis to group dune sites by presence/absence of plant species and correlate both clusters to validate the relationship between them, the salient aspects of this relationship were described and the spatial distribution of the groups was mapped.

Results

Eleven plant species were identified, six of them exclusive to the embryo dunes and the rest ubiquitous. The incipient development of these soils is reflected in a low content of organic matter, silt, clay, and moisture retention, with scattered data on granulometry, electrical conductivity, organic matter, and magnetite. Some significant correlations were found between some environmental properties, and the CCA showed a significant relationship between the presence of plant species and environmental variables (p-value of the Monte Carlo test = 0.026). The cluster analysis of dune sites according to environmental variables and the cluster analysis by presence/absence of plant species produces the formation of five groups of sites with significant environmental differences and five groups of sites with significant floristic differences. A significant connection (r = 0.471, p = 0.01) between the two clustering schemes also evidences the meaningful relationship between the presence of plant species and the environmental characteristics of the embryo dunes of Jalisco, Mexico. Differences in habitat preferences were observed among plant species exclusive to the embryo dunes; thus, Abronia maritima, Uniola pittieri, and Pectis arenaria showed a preference for embryo dunes with poor edaphic conditions, in contrast to Okenia hypogaea, Canavalia rosea, and Scaevola plumieri, which were mostly found in embryo dunes with higher fertility.

Introduction

Coastal dune ecosystems present a unique set of habitats with high environmental variability and specialized flora, which is rarely shared with other terrestrial ecosystems (Acosta, Carranza & Izzi, 2009; Ciccarelli, 2014; Angiolini, Bonari & Landi, 2018). These ecosystems present a natural stress gradient, perpendicular to the coastline, leading to compressed zoning of plant communities (Acosta et al., 2003; Forey et al., 2008; Carboni, Santoro & Acosta, 2011).

Influence of geomorphological aspects includes the elevation effect of dunes over sedimentation rate, soil sand content, and soil pH, which increase with higher elevations, whereas soil clay, fine silt, soil moisture, organic matter, total N, total K content, growth, and biodiversity of sand dune plants generally decrease with lower elevations (Pan, Zhang & Li, 2016).

Some characteristic properties that have been reported for primary dunes systems include the presence of a pH gradient, which decreases with distance from the shoreline; poor fertility conditions, and spatial variation in electrical conductivity (Kim & Yu, 2009; Willis et al., 2016). The absence of a relationship between soil chemical characteristics and plant community structure has also been reported (Willis et al., 2016).

Soil properties in coastal dunes determine important aspects of the vegetation establishment, functionality, and diversity. Soil organic matter determines plant biomass parameters, microorganisms, soil animals, and diversity of microarthropods; while pH establishes fundamental differences in the availability of phosphorus, thereby promoting vegetation strategies for the absorption of this element (Kooijman et al., 2020). Soil total organic carbon, total Nitrogen, Sodium, and total Phosphorus majorly determine bacterial communities structure (Gao et al., 2019).

Zuo et al. (2008), suggest that in a mobile dune, the distribution of plants is determined by a combination of edaphic properties and topographic aspects. In addition, Zuo et al. (2009) point out that the magnitude and degree of spatial heterogeneity in soil properties can influence the distribution pattern of plants and the plant succession of mobile dunes. These authors conclude that organic carbon, total nitrogen, electrical conductivity, pH, the slope of the land, very fine sand content, and water content in soil are the main factors that influence the presence and distribution of dune plants, and therefore diversity of vegetation. The distribution of dune plant species obeys the concordance between the spatial distribution of soil properties in dunes and environmental requirements from plant species. Diverse studies have shown this phenomenon, such as the research from Willis et al. (2016) who reported that Ipomoea pes-caprae and Spinifex littoreus were restricted to the foredunes, the leguminous forb Alysicarpus vaginalis and Perotis indica to the furthest sites from the strand, and Ischaemum indicum, a C4 perennial grass species, which was found to be ubiquitous to all sampled sites.

Dunes involve poorly developed soils, where vegetation plays a key role in the soil evolution process and the formation of microenvironments (Moreno-Casasola, 1982). The degree of stabilization of dune soils strongly depends on vegetation cover (Provoost, Jones & Edmondson, 2011). This is why many coastal foredune systems gain stabilization of the foredune morphology when vegetation cover is increased, moreover, since the fordune surface gets more stable, blowout events decrease (Martinez, Hesp & Gallego-Fernández, 2013; Schwarz, Brinkkemper & Ruessink, 2019). Vegetation contributes to the stabilization of dune soils by preventing erosion (Gao et al., 2019), promoting rainwater infiltration (Regüés et al., 2017), providing organic matter (Li et al., 2017), and contributing to a gradual improvement of soil structure (Torres-Guerrero et al., 2013).

The degree of stabilization of coastal dune soils gives rise to a classification in terms of mobile, semi-mobile, and stabilized zones; in mobile zones dune soils of the Gulf of Mexico, the first species to establish are tufted grasses and subshrub plants, which stop erosion and allow the accumulation of sand; they are considered the main initial fixers of the substrate (Martínez & Valverde, 1992). However, recently Biel, Hacker & Ruggiero (2019) concluded that dune-forming species also include annual and perennial forbs with prostrate growth patterns and species that primarily propagate by seed, rhizomes, and stolons; furthermore, they found that sand accretion in dunes is strongly influenced by patterns of vegetation growth.

Coastal dunes are key ecosystems, with particular geomorphological and sedimentary characteristics, which provide extremely important environmental services, such as coping with rising sea levels to prevent impacts inland; it is estimated that in México there is approximately 800,000 ha of coastal dunes, distributed in around 80% of the Mexican coastline (Jiménez-Orocio, Espejel & Martínez, 2015). Coastal dune vegetation and soils have been studied more in the Gulf of México than in the Pacific Ocean coastline; therefore, most of the soils and vegetation of this region remain unstudied, which limits the full understanding of its formation and evolution process.

The objective of this research was to identify the edaphic and geomorphological properties that majorly determine the presence of plant species in the coastal embryo dunes of Jalisco, México. Embryo dunes are often ephemeral but can develop to become established coastal foredunes (Montreuil et al., 2013b). Embryo dunes presence and extension depend on beach width, storm intensity and frequency, growing season precipitation, and sand nourishment (van Puijenbroek et al., 2017). Severe storms surge events cause areas of erosion between dunes, aligned with the dominant wind direction, yielding a sediment volume reduction in the embryo dune; as a counterpart, an increase in sediment volume occur because of seaward accretion due to onshore aeolian sediment supply as opposed to either a gain in height or an expansion of the dune field in an alongshore direction (Montreuil, Bullard & Chandler, 2013a). Commonly, embryo dunes are populated by pioneer vegetation species that can cope with sand burial (Gómez et al., 2017). Initially, embryonic dunes may increase species richness, but later they may compete for space, further reducing species richness (van Puijenbroek et al., 2021).

Materials and Methods

Study area

Along 358 km of the coastline of Jalisco, México, and covering the municipalities of Puerto Vallarta, Cabo Corrientes, Tomatlán, La Huerta, and Cihuatlán, we can find different coastlines, mainly: Sandy beaches, cliffs, mangroves, and estuaries. The sandy beaches occupy approximately 181 km (50.5%) in the State and are formed by a beach followed by sand mounds or dunes.

In the embryo dunes, 29 sites were sampled: (1) Las Playitas, (2) North Aquiles Serdán, (3) Los Naranjos, (4) Aquiles Serdán, (5) Mayto, (6) Las Villas, (7) El Realito, (8) El Coco, (9) Las Peñitas (San Carlos), (10) Playón de Mismaloya, (11) North La Gloria, (12) South La Gloria, (13) La Limonera, (14) Las Peñitas, (15) North Chalacatepec, (16) South Chalacatepec, (17) La Soledad, (18) Chachalacas, (19) North Chamela, (20) South Chamela, (21) Playa Chica, (22) North El Tecuán, (23) South El Tecuán, (24) North Tenacatita, (25) South Tenacatita, (26) Punta Tenacatita, (27) Melaque (Laguna del Tule), (28) Isla Navidad, and (29) Rio Marabasco. These sites are located between latitudes 19.16° and 20.21° north and longitudes 104.61° and 105.41° west (Fig. 1).

Figure 1 Location of sample sites.

Map data © 2021 Esri, Garmin, GEBCO, NOAA NGDC, INEGI.

Geomorphology classifies the study area as rocky coasts, where stretches of cliffs and rocky points alternate and are bordered by bays and thin beaches. With 62 km in length, Mismaloya beach is the one with the greatest continuity (Secretaría de Medio Ambiente y Recursos Naturales, 2013). The climate is predominantly warm semi-arid and warm sub-humid, with a mean annual temperature between 22 °C and 26 °C and mean annual precipitation between 400 and 800 mm. Urban and tourist use of soils represents 4% of the surface, while natural vegetation covers 96% of the territory. Nevertheless, vegetation is currently strongly impacted by extensive livestock (Secretaría de Medio Ambiente y Recursos Naturales, 2013).

Pre-field work

A geographic information system (GIS) was integrated by using the ArcGIS Desktop 10.3 software and with national open data that include the topics hydrological basins (INEGI, 2001b), land use and vegetation (INEGI, 2017b), types of soils (INEGI, 2017a), types of rocks (INEGI, 1999); communication routes, urban areas, streams, water bodies, place names and contour lines (INEGI, 1998); and climate types (INEGI, 2001a).

A digital elevation model and a shading model were generated from the contour lines (INEGI, 1998) in the ArcGIS system, with a pixel size of 20 m. Both models were integrated into the GIS. False-color compositions with bands 3,2,1 were also generated in ArcGIS with 19 Spot-4 satellite images, with a spatial resolution of 10 m mono-spectral pixel, and 20 m multispectral pixel. The spectral resolution for band 1 (Green) is 0.50-0.59 µm; for band 2 (red) is 0.61–0.68 µm; for band 3 (near-infrared) is 0.78–0.89 µm, and for band 4 (mid-infrared) is 1.58–1.75 µm. The single-shot date was between January and February 2015.

Finally, relief forms were delimited and classified by using the false-color compound according to the modified physiographic method for México (Ortiz-Solorio & Cuanalo de la Cerda, 1984).

Fieldwork

Soil sampling was made from January to March 2017, under a random stratified design, considering 1–2 sites per each interpreted topoform. Soil samples were collected 10 m away from the starting point of the dune, in a perpendicular direction and opposite to the beach. A 30 × 30 × 30 cm hole was made in the ground, subsequently, a representative sample of soil depth was taken from the wall of the well that faces the sea; the vegetation island effect was avoided in every case.

Data collection in each observation site included: geographical location (UTM coordinates, zone 13N), altitude (masl), vegetation coverage (%), description of the topographic shape, terrain slope (%), slope orientation, distance to the sea, identification of vegetation and land use according to the classification system used by the Mexican government for national cartography (INEGI, 2017c); phytosociological samplings according to the methodology of Braun-Blanquet (1979) and identification of flora according to Macías-Rodríguez et al. (2019).

Laboratory

Prior to laboratory determinations, samples collected were air-dried and labeled. Lab determinations included wet and dry color with Munsell Tables, particles size (% sand) according to the American Classification System (USDA, 2017), organic matter (%) with the Walkley-Black procedure (van Reeuwijk, 2002), pH in solution 2:1, measured with a potentiometer (van Reeuwijk, 2002), electrical conductivity (dSm−1) in saturation extract, measured with a conductivity meter (van Reeuwijk, 2002), magnetite content (%), by separating it with a magnet, moisture retention (%) with the gravimetric method (van Reeuwijk, 2002).

Metric variables totalized 17; 12 edaphic: moisture retention, organic matter, electrical conductivity, pH, magnetite, gravel, very coarse sand, coarse sand, medium sand, fine sand, very fine sand, fraction finer tan very fine sand; one geographic: latitudinal gradient; two geomorphological: distance to the sea, topoform altitude; and two vegetation variables: vegetation coverage and number of plant species. The parametric variables included the color of sands in wet and dry and the orientation of the topographic shape. As complementary data, rock types and climate types were included. Finally, a table of presence/absence of plant species was integrated from species identified with the phytosociological method.

The variable latitudinal gradient was considered quantitatively not by latitudinal values per se but by ordinal values from 1 to 29 (according to the number of sample sites), assigning the value 1 to the northernmost site and 29 to the southernmost site.

Statistical and spatial analysis

For the statistical analysis, we first explored how the environmental variables are correlated with each other, using Spearman’s r.

Subsequently, to identify the relationships between the environmental variables (including edaphic, geomorphological, and geographic), we used multivariate statistics; thus, two principal component analyzes (PCA) were made, the first with all the variables, to generate the matrix of anti-image correlations and to be able to select the variables with the best measures of sample adequacy, in the second PCA the number of participating variables and the number of eigenvectors was reduced, yielding an increase in the variance explained in the first two axes. Further, a cluster analysis was performed to group the sample sites to facilitate the identification and interpretation of the environments explored; to validate the congruence of the groups, an analysis of variance (ANOVA) was carried out.

The PCA and its canonical form, the redundancy analysis (RDA), are more robust than the canonical correspondence analysis (CCA) to analyze the interrelationships between the attributes of the soil and the shape of the terrain, since their behavior, in general, is more linear (Odeh, Chittleborough & McBratney, 1991; Park & Burt, 2002; Kim & Yu, 2009). However, CCA is frequently recommended to analyze the relation between floristic data and environment (for example, Kim & Yu, 2009, in dune system). Between PCA and RDA, we decided to use PCA because, in this first stage of our project, it was not intended to distinguish between edaphic and geomorphological variables, but to directly know how environmental variables influence the presence of plant species.

The relationship between environmental variables and plant species was studied with two separate methods, first, a CCA was made to define the environment-plant relationship in the study area, and identify the proximity of the environmental variables with the plant species. On the other hand, hierarchical conglomerate analysis was made based on the presence/absence of plant species. Then, we proceeded to correlate (Pearson’s r) this grouping with the environmental grouping previously obtained to validate the relationship between both cluster schemes. Once the relationship between the two groups had been validated, outstanding aspects of this relationship were described and the spatial distribution of the groups was represented on a map.

The software used to perform these analyzes was IBM SPSS Statistics 19 and XLSTAT 2021.3.1.

Results

Description of plant species and environment of the study area

From the 29 sampled sites in embryo dunes, 11 species of plants were identified, five of them ubiquitous (Fig. S1), this is present not only in the embryo dunes (Fig. 2) but also in more consolidated dunes, the remaining six species were present only in the embryo dunes (Table 1), and not consistently since in seven of the 29 sites were not present these species.

Figure 2 Exclusive plant species of embryo dunes in the study area.

(A) Abronia maritima, (B) Canavalia rosea, (C) Okenia hypogaea, (D) Pectis arenaria, (E) Scaevola plumieri, (F) Uniola pittieri. Photo credit: Miguel Ángel Macías-Rodríguez.

Table 1 Plant species identified, growth pattern, and types of dunes where they are present.

Species	Growth pattern	Distribution	Sites where it is present	
Prosopis juliflora	Shrub	Ubiquitous	1,10,16,17,22,24	
Canavalia rosea	Climbing plant	Exclusive to embryo dunes	16,17,18,21	
Abronia maritima	Herb	Exclusive to embryo dunes	9,12,15	
Acalypha monostachya	Herb	Ubiquitous	5,22,24,28	
Ipomoea pes-capreae	Herb	Ubiquitous	2,3,4,8,9,10,11,12,13,14,16,17,
18,19,21,22,23,25,26,27,28,29	
Okenia hypogaea	Herb	Exclusive to embryo dunes	5,8,16,19,20,26,27,28,29	
Pectis arenaria	Herb	Exclusive to embryo dunes	5,7,9,10,12,13,18,19,21,22,25	
Scaevola plumieri	Herb	Exclusive to embryo dunes	11,12,13,16	
Distichlis spicata	Grass	Ubiquitous	7,11,12,13,19,22,24,28	
Jouvea pilosa	Grass	Ubiquitous	2,3,4,5,6,8,9,13,16,17,20,21,22,
23,24,26,27,28	
Uniola pittieri	Grass	Exclusive to embryo dunes	14,15,18	

Ipomoea pes-capreae and Jouvea pilosa were the most common species (22 and 18 sites, respectively), followed by Pectis arenaria and Okenia hypogaea (11 and 9 sites, respectively). The less frequent species were Abronia maritima and Uniola pittieri, since they only occurred in three sites. The average frequency per site of ubiquitous species and of exclusive species of embryo dunes was 2.0 and 1.2, respectively.

Regarding the environment of the sites sampled, as expected, the incipient development of soils in embryo dunes reflects a low content of organic matter, silt and clay, and moisture retention. Furthermore, as shown in Table 2, the diversity of the environments is reflected in the dispersed data of the granulometry (gravel, very coarse, very fine, fine, and, fraction finer than very fine sand), and other edaphic properties such as electrical conductivity, organic matter, and magnetite. Non-edaphic variables are less dispersed than the edaphic ones.

Table 2 Descriptive statistics of environmental variables in embryo dunes.

Variable	N	Minimum	Maximum	Range	Mean	Standard deviation	Coefficient of variation	
Moisture retention (%)	29	1.2	6.4	5.2	3.1	1.481	0.48	
Organic matter (%)	29	0.0000	0.2725	0.2725	0.0718	0.065	0.91	
Electrical conductivity (dSm−1)	29	0.05	4.04	3.99	0.88	1.036	1.18	
pH	29	6.23	8.66	2.43	7.36	0.629	0.09	
Magnetite (%)	29	1.4	33.8	32.4	11.7	9.180	0.78	
Gravel (%)	29	0.0	3.2	3.2	0.4	0.740	1.87	
Very coarse sand (%)	29	0.0	71.5	71.5	11.3	18.843	1.67	
Coarse sand (%)	29	0.0	75.1	75.1	30.6	23.627	0.77	
Medium sand (%)	29	2.9	90.9	88.0	52.1	29.592	0.57	
Fine sand (%)	29	0.0	18.8	18.8	5.0	4.415	0.88	
Very fine sand (%)	29	0.0	1.4	1.4	0.3	0.333	1.27	
Fraction finer tan very fine sand	29	0.0	1.0	1.0	0.3	0.266	0.87	
Latitudinal gradient	29	1.0	29.0	28.0	15.0	8.515	0.57	
Distance to sea (m)	29	35.7	173.8	138.0	72.5	28.998	0.40	
Topoform altitude(m)	29	5.0	22.0	17.0	11.2	4.473	0.40	
Vegetation coverage (%)	29	5.0	70.0	65.0	30.0	21.547	0.72	
Number of plant species	29	1.0	6.0	5.0	3.2	1.391	0.44	

The sample sites are distributed in three dune systems, (a) beaches and dunes dissected by cliffs, staggered, below the continental level, an example of this are the sites Las Playitas and La Soledad (Fig. S2); (b) beaches and dunes of great length, which forms high dunes and above the continental level, tends to form depressions, as is the case of the El Coco and La Limonera sites; (c) beaches and dunes of variable length, tend to form beaches and wide dunes, sometimes in bar form, which derives depressions, as seen in Fig. S2, North Chamela and South El Tecuán sites.

Relationships between environmental variables

Vegetation coverage correlated significantly with topographic, edaphic, and geographic variables (Table 3). Higher correlations were obtained with pH (r = 0.690, p = 0.001), fine sand (r = 0.600, p = 0.001), magnetite (0.644, p = 0.001), and latitudinal gradient (r = 0.809, p = 0.001) showing that a major presence of vegetation should be expected southward the study area. Latitudinal gradient also strongly correlated with pH (r = 0.834, p = 0.001) and with magnetite (r = 0.731, p = 0.001), and with a moderate but significant coefficient with fine sand (r = 0.670, p = 0.001). These correlations also indicate a geographical pattern in the spatial distribution of these edaphic variables. Significant and moderately high correlations were also obtained between soil properties: pH with magnetite (r = 0.639, p = 0.001), pH with fine sand (r = 0.662, p = 0.001), moisture retention with very fine sand (r = 0.613, p = 0.001). The rest of the significant correlations can be seen in Table 3.

Table 3 Correlation coefficients between environmental variables.

	Organic matter	Electrical conductivity	pH	Very coarse sand	Fine sand	Magnetite	Altitude of the topoform	Vegetation coverage	Distance to the sea	Moisture retention	Orientation of the topoform	
Latitudinal gradient	r	−0.324	−0.187	0.834	−0.466	0.670	0.731	−0.522	0.809	−0.329	0.285	−0.318	
p	0.087	0.332	0.000	0.011	0.000	0.000	0.004	0.001	0.081	0.134	0.093	
Organic matter	r		−0.062	−0.332	0.114	−0.108	−0.190	0.468	−0.455	−0.040	0.201	−0.155	
p		0.749	0.079	0.556	0.578	0.324	0.010	0.013	0.837	0.296	0.423	
Electrical conductivity	r			−0.242	0.121	−0.281	−0.105	0.014	−0.092	0.185	−0.143	−0.277	
p			0.205	0.533	0.139	0.586	0.941	0.636	0.336	0.458	0.146	
pH	r				−0.480	0.662	0.639	−0.472	0.690	−0.310	0.241	0.334	
p				0.008	0.000	0.000	0.010	0.001	0.102	0.207	0.076	
Very coarse sand	r					−0.682	−0.330	0.199	−0.519	−0.394	−0.540	−0.400	
p					0.000	0.080	0.301	0.004	0.034	0.002	0.032	
Fine sand	r						0.419	−0.247	0.600	−0.016	0.513	0.547	
p						0.024	0.196	0.001	0.934	0.004	0.002	
Magnetite	r							−0.454	0.644	−0.195	0.241	0.214	
p							0.013	0.000	0.310	0.209	0.264	
Altitude of the topoform	r								−0.559	0.315	−0.123	−0.175	
p								0.002	0.096	0.526	0.363	
Vegetation coverage	r									−0.035	0.224	0.415	
p									0.857	0.243	0.025	
Distance to the sea	r										−0.040	0.072	
p										0.835	0.711	
Moisture retention	r											0.403	
p											0.030	
Notes:

r, Spearman’s rho correlation coefficient; p, p-value.

All bold values are significant at the level p < 0.05.

Grouping of embryo dunes according to environmental variability

As a by-product of the PCA, the anti-image correlation matrix allowed obtaining the Kaiser–Meyer–Olkin (KMO) sampling adequacy measures. The criterion to keep variables was KMO > 0.45, therefore distance to the sea (KMO = 0.175), number of plant species (KMO = 0.255), fraction finer than very fine sand (KMO = 0.307), and gravel (KMO = 0.372) were excluded from the analysis. For the PCA, nine variables were considered: moisture retention (KMO = 0.583), organic matter (KMO = 0.450), pH (KMO = 0.639), magnetite (KMO = 0.613), very coarse sand (KMO = 0.677), fine sand (KMO = 0.847), latitudinal gradient (KMO = 0.737), altitude of the topoform (KMO = 0.745), and vegetation coverage (KMO = 0.705). According to the Kaiser criterion, we selected the first two components (eigenvalue >1). The first component explains 52.34% of the variance and the second component 17.84%, accumulating 70.18% of the total variance. The first component is determined by latitudinal gradient, vegetation coverage, pH, fine sand, magnetite, and inversely by very coarse sand and altitude of the topoform. The second component was determined by moisture retention and organic matter. Table 4 shows the eigenvectors of the final nine environmental variables from the first two principal components.

Table 4 Principal component analysis and cluster analysis statistics.

	Principal component 1	Principal component 2	
Variance explained	
% of Variance	52.340	17.838	
Cumulative %	52.340	70.178	
Eigenvectors	
Latitudinal gradient	0.915	−0.087	
Organic matter	−0.401	0.679	
pH	0.867	−0.072	
Very coarse sand	−0.669	−0.469	
Fine sand	0.781	0.399	
Magnetite	0.753	−0.112	
Moisture retention	0.431	0.712	
Vegetation coverage	0.879	−0.181	
Altitude of the topoform	−0.616	0.446	
ANOVA	
Between Groups	
Sum of squares	23.116	23.035	
df	4	4	
Mean Square	5.779	5.759	
F	28.396	27.836	
Sig.	0.001	0.001	
Within Groups	
Sum of squares	4.884	4.965	
df	24	24	
Mean Square	0.204	0.207	

Hierarchical cluster analysis with squared Euclidean distance <10, permitted to identify four groups and an atypical site. Group 1 includes nine sites; Groups 3 and 5 are integrated by eight sites each; Group 4 contains three sites, and Group 2 includes only the atypical site (Fig. S3). The scatter plot (Fig. 3) generated with the scores of the two first components, shows the distribution and proximity between groups identified in the cluster analysis. There is a statistically significant difference between groups according to the two scores, as shown by the one-way ANOVA (Table 4); F (4.24) = 28.396, p = 0.001 for the first score, and F (4.24) = 27.836, p = 0.001 for the second score.

Figure 3 Scatter plot constructed with the PCA component scores.

The numbers framed within the plot correspond to the site number.

Description of the groups from environmental cluster analysis

According to dry color values (pale brown and very pale brown), the sands of Group 1 sites are mostly made up of granite and conglomerate granite (88.9% of sites). This group presents the lowest mean values of moisture retention, pH, magnetite, fine sand, vegetation coverage, the number of plant species, and the highest mean values of very coarse sand and altitude of topoform (Table 5).

Table 5 Environmental characteristics and number of plant species for the groups from the cluster analysis.

Variable	Group 1 (n = 9)	Group 2 (n = 1)	Group 3 (n = 8)	Group 4 (n = 3)	Group 5 (n = 8)	
x¯	s	x¯	x¯	s	x¯	s	x¯	s	
Moisture retention	1.94	0.98	5.97	4.13	0.99	1.90	0.72	3.39	1.30	
Organic matter	0.07	0.04	0.21	0.12	0.07	0.02	0.03	0.03	0.03	
Electrical conductivity	1.02	0.82	0.08	0.91	1.11	1.06	0.96	0.73	1.37	
pH	6.78	0.33	6.67	7.40	0.47	7.74	0.83	7.92	0.34	
Magnetite	6.47	9.00	4.00	11.54	8.44	22.07	10.29	14.94	6.43	
Very coarse sand	28.33	24.31	1.40	1.35	1.73	15.93	20.32	1.62	2.78	
Fine sand	1.38	1.82	3.80	6.12	3.06	2.27	2.57	9.20	4.59	
Distance to the sea	70.73	19.19	82.35	70.79	24.25	92.98	71.78	67.22	25.48	
Topoform altitude	14.89	5.62	15.00	11.00	1.41	6.67	1.53	8.38	1.92	
Vegetation coverage	10.00	5.59	10.00	27.50	21.71	56.67	5.77	47.50	10.35	
Number of plant species	3.22	1.39	2.00	3.25	1.28	3.67	2.08	4.50	1.41	
Note:

x¯, mean; s, standard deviation; n, number of observations (sites).

In Group 2, which is integrated only by the atypical site, the source material is granite, light yellowish-brown color (dry color), with 5.9% of moisture retention, organic matter 0.202%, pH 6.67, magnetite 4%, vegetation coverage 10% and two plant species.

Group 3 sites have a mineral fraction formed from sandstone-conglomerate (50%) and granite (25%). The dry sands are light brownish gray (50%) and light yellowish brown (25%). As can be seen in Table 5, this group has the lowest mean value in very coarse sand and the highest mean value for moisture retention and organic matter.

In Group 4, the sites present a mineral fraction formed from volcanoclastics (100%), and the sands are dark gray (66.7%) when wet. In Table 5, it can be observed that this group presents the lowest mean value in organic matter and the highest mean values of electrical conductivity, magnetite, distance to the sea, and vegetation coverage.

The mineral fraction in sites of Group 5 is formed mainly from volcanoclastics (37.5%) and acid tuff (25%), the color of sands in wet are brown, greyish brown, and very dark greyish brown (75%). This group has the lowest mean value of electrical conductivity and distance to the sea, and the highest mean values of pH, fine sand, and the number of species.

Relationship between environmental variables and plant species

Canonical correspondence analysis resulted significant (p-value Montecarlo test = 0.026, 500 permutations) in the first two Axes, showing that vegetation data are related to the environmental properties data. As can be seen in Table 6, the cumulative percentage variance of species-environment relation totalizes 54.01 in the first two axes.

Table 6 Summary of canonical correspondence analysis for the first two axes at the embryo dunes of the Pacific Coast in Jalisco, México.

	Axis 1	Axis 2	
Eigenvalue	0.427	0.212	
Cumulative percentage variance of species-environment relation	36.08	54.01	
p-value Monte Carlo Test	0.026	0.026	
Correlations with individual environmental attributes	
Latitudinal gradient	0.084	0.061	
Organic matter	−0.139	−0.371	
pH	0.011	−0.026	
Electrical conductivity	0.708	0.236	
Very coarse sand	−0.126	0.391	
Fine sand	0.106	0.103	
Magnetite	−0.106	−0.195	
Altitude of the topoform	0.045	−0.198	
Vegetation coverage	−0.064	0.118	
Distance to the sea	0.272	0.079	
Moisture retention	0.262	−0.002	
Orientation of the topoform	−0.088	0.456	
Note:

All bold values are significant at the level, p < 0.05.

The electrical conductivity was highly and significantly correlated (0.708) with Axis 1, while organic matter (−0.371), very coarse sand (0.391), and orientation of the topoform (0.456) correlated significantly, although with a low-value coefficient with the Axis 2.

The CCA ordination plot (Fig. 4) allows visualizing the proximity of the species with each environmental variable. Thus, the exclusive species of embryo dunes show to be the most distant from the environmental variables; good examples of this condition are Abronia maritima and Uniola pittieri (Fig. 4). However, according to the plot, the variable that can best explain the presence of Okenia hypogaea is organic matter, while the presence of Pectis arenaria is more linked to fine sand. In the case of Canavalia rosea, is the influence of fine sand, pH, and distance to the sea, that favors the presence of this species (Fig. 4).

Figure 4 Ordination diagram that illustrates the relationships between environmental properties (lines) and plant species (dots).

Species: Abr: Abronia maritima, Aca: Acalypha monostachya, Can: Canavalia rosea, Dis: Distichlis spicata, Ipo: Ipomoea pes-capreae, Jou: Jouvea pilosa, Oke: Okenia hypogaea, Pec: Pectis arenaria, Pro: Prosopis juliflora, Sca: Scaevola plumieri, Uni: Uniola pittieri. Environment variables: alt: Altitude of the topoform, cob: Vegetation coverage, dis: Distance to the sea, lat: Latitudinal gradient, ori: Orientation of the topoform, af: Fine sand, amg: Very coarse sand, ce: Electrical conductivity, hum: Moisture retention, mg: Magnetite, mo: Organic matter, ph: pH.

Regarding ubiquitous species, they show more interdependence with environmental variables, thereby organic matter is the most explanatory variable for the presence of Acalypha monostachya, Distichlis spicata, and Jouvea pilosa. Ipomoea pes-capreae is related to retention of moisture, pH, and electrical conductivity. Prosopis juliflora is the only ubiquitous species that is not related to any of the environmental variables.

Table 7 shows the simple correlation of plant species with some environmental variables. It is worth mentioning that this table only includes five of the 12 variables considered in the CCA, since for the remaining seven variables no significant correlation was found (p < 0.05) with any plant species. It stands out that two exclusive species of embryo dunes show significant correlations; Okenia hypogaea with two non-edaphic environmental variables (Latitudinal gradient and Distance to the sea) and Uniola pittieri with two edaphic variables (Electrical conductivity and very Coarse sand). The other four exclusive species of embryo dunes did not show any significant correlation with environmental variables.

Table 7 Correlation coefficients between plant species and some environmental variables.

	Latitudinal gradient	Electrical conductivity	Very coarse sand	Distance to the sea	Orientation of the topoform	
Prosopis juliflora	r	0.000	0.122	0.219	−0.224	0.106	
p	1.000	0.528	0.254	0.243	0.586	
Canavalia rosea	r	0.143	0.227	−0.281	0.287	0.276	
p	0.458	0.236	0.139	0.131	0.148	
Abronia maritima	r	−0.122	0.230	−0.163	0.189	−0.320	
p	0.529	0.230	0.399	0.325	0.091	
Acalypha monostachya	r	0.227	−0.490	−0.042	−0.227	0.110	
p	0.236	0.007	0.829	0.236	0.569	
Ipomoea pes-capreae	r	0.260	−0.048	−0.227	0.202	0.383	
p	0.173	0.804	0.237	0.293	0.040	
Okenia hypogaea	r	0.383	−0.143	0.232	−0.401	−0.262	
p	0.040	0.461	0.226	0.031	0.170	
Pectis arenaria	r	−0.034	−0.093	−0.094	−0.034	0.304	
p	0.861	0.630	0.629	0.861	0.109	
Scaevola plumieri	r	−0.096	−0.060	−0.168	0.143	0.138	
p	0.622	0.758	0.385	0.458	0.476	
Distichlis spicata	r	0.148	−0.323	−0.319	−0.138	0.319	
p	0.445	0.088	0.092	0.474	0.092	
Jouvea pilosa	r	0.034	−0.365	0.179	0.068	−0.088	
p	0.861	0.050	0.354	0.726	0.649	
Uniola pittieri	r	0.027	0.514	−0.407	0.271	−0.101	
p	0.889	0.004	0.029	0.156	0.601	
Notes:

r, Spearman’s rho correlation coefficient; p, p-value.

All bold values are significant at the level p < 0.05.

Regarding the ubiquitous species, three showed significant correlation with at least one environmental variable, Acalypha monostachya, and Jouvea pilosa, in which electrical conductivity influences the sites that these species colonize. Meanwhile, Ipomoea pes-capreae is positively influenced by the orientation of the topoform, that is, the south orientation of the topoform is favorable for it.

Grouping of embryo dunes according to plant species

Hierarchical cluster analysis with squared Euclidean distance, allowed us to identify five groups of plant species. Group 1 is composed of nine sites, Groups 2 and 3 including seven sites each, and Groups 4 and 5 integrated by three sites each (Fig. S4).

As can be seen in Table 8, Group 1 is the only one that does not have a plant species that is present in all sites, it is also the group that reports the fewest species, Group 2 is dominated by the presence of Pectis arenaria; Group 3 by Okenia hypogaea; Group 4 by Scaevola plumieri, and Group 5 by Canavalia rosea. Abronia maritima is scarcely present in Groups 1, 2, and 4, while Uniola pittieri is present in Groups 1 and 5 but with little representation.

Table 8 Plant species that determined the conformation of the groups in cluster analysis.

Group	n	Species	Sites with presence	Sites %	
1	9	Abronia maritima	1	11.1	
Uniola pittieri	2	22.2	
2	7	Abronia maritima	1	14.3	
Okenia hypogaea	2	28.6	
Pectis arenaria	7	100.0	
3	7	Canavalia rosea	1	14.3	
Okenia hypogaea	7	100.0	
Scaevola plumieri	1	14.3	
4	3	Abronia maritima	1	33.3	
Pectis arenaria	2	66.7	
Scaevola plumieri	3	100.0	
5	3	Canavalia rosea	3	100.0	
Pectis arenaria	2	66.7	
Uniola pittieri	1	33.3	
Note:

n, number of observations (sites).

Relationship between environmental and plant species clusters

Grouping of embryo dunes by environmental variables (GVA) and by presence/absence of plant species (GEF) have a significant correlation (Pearson r = 0.471, p = 0.01), which shows that there is a relationship between environmental variables and plant species. Figures 5 and 6 show the geographic distributions of both clusters.

Figure 5 Geographic distribution of groups of sites according to environmental variables (GVA).

Map data © 2021 Esri, Garmin, GEBCO, NOAA NGDC, INEGI.

Figure 6 Geographic distribution of groups of sites according to the presence of plant species (GEF).

Map data © 2021 Esri, Garmin, GEBCO, NOAA NGDC, INEGI.

Five out of nine sites of GEF 1 (with the least presence of species) are related to GVA 1 and GVA 2; these groups contain the dunes with the thickest values of particles and the lowest values in the parameters that indicate fertility. These groups are found mostly in the northern portion of the study area, on the open coastline, and with few dune cords. A similar condition is found in GEF 2 (group predominated by Pectis arenaria); four of the seven sites coincide with GVA 1.

The Group GEF 3 (predominated by Okenia hypogaea) has a dispersed distribution and is related to all the GVA (1, 3, 4, 5). GEF 3 is formed by dunes with the highest vegetation coverage and the highest magnetite contents, it is found mainly in the southern portion of the study area, in bays with wide beaches and numerous dune ridges.

Scaevola plumieri predominates in GEF 4, with a distribution in the central part of the territory studied and it is completely contained in GVA 3. This group shows the dunes with the highest content of organic matter as well as moisture retention capacity; it is present on the open coastline associated with lagoon systems and with numerous dune ridges. Finally, GEF 5 (with a major presence of Canavalia rosea) has a distribution in the central part of the study area and it is contained totally in GVA 5. This group has the highest pH, the finest sand grains, and the greater presence of species, it is present in coastlines protected by bays with narrow beaches and few dune ridges.

Discussion

The high values of coefficient of variation found in diverse edaphic variables, evidence soil properties that still keep great dynamics, which is characteristic for not consolidated soils in embryo dunes (van Puijenbroek et al., 2021). In such edaphic conditions, the establishment of vegetation occurs with pioneer species or with species with a certain level of rusticity or adaptation to poor soils, which limits the number of species present (Ripley & Pammenter, 2008; Šile et al., 2017). However, the plant communities that these species usually form are very diverse in composition (species) and coverage, which is also due to the spatial variation of edaphic properties (Table 1).

Environmental diversity

By the results obtained in the correlation analyses, the organic matter in the embryo dunes is azonal, therefore, it is directly related to topoform altitude (both determined by wind processes), and is inversely related to vegetation coverage. Through the transport of sea salt and sand grains, wind processes provide materials that contain various organic and inorganic components, they are the most important source of nutrients for the plant and enrichment of organic matter in the dunes (Yu, Rhew & Kim, 2004).

Correlation analysis also reveals that when pH, fine sand, and magnetite are increased, greater vegetation coverage is favored. Moreover, pH, fine sand, and magnetite are negatively correlated with topoform altitude, thus vegetation coverage is expected to be greater in lower topoform altitudes. Similarly, Kim & Yu (2009) identified pH, Ca2+, and Mg2+ as explanatory variables for the presence of vegetation, and found that altitude, distance to the sea, and aspect of the slope are negatively correlated with vegetation.

Regarding the relationship soil and vegetation have, the soil must be understood as the set of its properties and not only its properties independent of each other (Hironaka, Fosberg & Neiman, 1991). This is why soils characterization is preferably carried out with multivariate techniques. (Kim, Yu & Park, 2008). With the grouping of sites with cluster analysis PCA, it is observed that GVA 1 and GVA 2 contrast in fertility concerning GVA 3, GVA 4, and GVA 5. While GVA 1 and GVA 2 are located in more northern and less fertile sites; GVA 3, GVA 4, and GVA 5 are distributed in more southern and more fertile places. According to the correlation analysis, this condition does not depend on the moisture retention index, nor the content of organic matter, as might be supposed, but on the geological material that gave rise to the sands, which determine their granulometry, pH, and content of magnetite (Weil & Brady, 2017).

In the study area, geological materials most susceptible to physical weathering (plutonic) are found in the northern portion, while those susceptible the most to chemical weathering (volcanic) are located in the southern portion, and in the central region, there is a combination of both geological materials; hence the correlation analysis of the latitudinal gradient reflects the material that gave rise to the dune sands. Consequently, pH, magnetite, and fine sand increase from north to south, and coarse sand decreases. Volcanic materials produce sands with a greater capacity to support plants (evidenced by the percentage of vegetation coverage and number of plant species) than plutonic materials (Weil & Brady, 2017). On the other hand, the hardness of geological materials and their age of exposure to weathering strongly influences the relief conformation. Young materials, such as the plutonic rocks in the north of the study area, determine abrupt reliefs, while older materials with less hardness, like the volcanic rocks in the southern portion of the region, produce reliefs with gentler slopes (Servicio Geológico Mexicano, 2017).

Relationship between environmental variables and plant species

Variance explained in the first two axes from the CCA is relatively low in comparison to other studies (Kim & Yu, 2009) but sufficient to demonstrate the existence of a relationship between environment and plant species in the ecosystems of embryo dunes. This can be explained by considering the nature of the present investigation, which considered sampling soils in a great territorial extension (29 sites distributed in 181 km of littoral); besides, samples were taken in incipient soils, where edaphic properties are just developing (Miller, Gornish & Buckley, 2010).

In general, the plant species exclusive of embryo dunes are little dependent on environmental properties, because they are the pioneer species to colonize the soils in their early stages of formation, and have physiological adaptations that allow them to grow in a difficult environment, such as prostrate habit to avoid strong winds, deep tap roots to obtain available moisture, waxy leaves to retain moisture and withstand winds, and high salt tolerance (Ripley & Pammenter, 2008). Abronia maritima y Uniola pittieri showed to be very good examples of this condition. The other four exclusive species of embryo dunes show an incipient edaphic dependence, but it is not clear how this relationship occurs.

Plant species exclusive of embryo dunes play a key role in stabilizing the dune, and the accumulation of finer particles such as fine and very fine sand, silts, organic matter, and salt, which are carried by the wind processes, which constitute the most important source of nutrients for the plant and enrichment of organic matter in the dunes (Yu, Rhew & Kim, 2004). As this process of sedimentation of fine particles occurs, suitable conditions are generated for the ubiquitous species to arrive (Pan, Zhang & Li, 2016; Wang et al., 2019; van Puijenbroek et al., 2021).

Cluster analysis by presence/absence of plant species reveals that some species have a predilection for poorer edaphic conditions, others instead prefer more developed edaphic conditions. Abronia maritima, Uniola pittieri, and Pectis arenaria tend to be distributed in embryo dunes with thicker sands and less edaphic development, but not Okenia hypogaea, Canavalia rosea, and Scaevola plumieri, which tend to predominate in embryo dunes with higher content of magnetite, organic matter, and fine sands, as well as higher values of pH and moisture retention. Angiolini, Bonari & Landi (2018) mention that pioneer species of embryo dunes show a maximum probability of occurrence at low values of total organic carbon, while the ubiquitous species show a contrary trend.

The complex interrelations between geomorphology, soil, and vegetation are key factors in the formation of ecological patterns and processes (Landi, Ricceri & Angiolini, 2012). This complexity can be understood as a chain of causal connections between geomorphological phenomena, soil conditions, and vegetation patterns (Moreno-Casasola, 1982; Kim & Yu, 2009; van Puijenbroek et al., 2021). The embryo dunes of the coast of Jalisco, México, show a complex interaction between plants and ambient, which is validated by the relatively low but significant explanation of the variance with the CCA and the moderately low but significant correlation between environmental grouping and plant species grouping. Similar results have been obtained by other authors, who explain them in terms of the complexity of the interactions that occur within the coastal ecosystems (Ciccarelli, 2014; Ruocco et al., 2014; Angiolini, Bonari & Landi, 2018).

Conclusions

The embryo dunes of the Pacific coast in the state of Jalisco, Mexico, are incipient soils with variable parameters, which limit the number of plant species that can be established, but favor plant communities of variable composition, which in turn generate diverse landscapes.

Despite the youth of the soils of the embryo dunes, three edaphic variables were observed to be closely related: pH, fine sand, and magnetite, which in turn are related to the vegetation cover, thus, these variables are the main ones responsible for variability in soil fertility. Although for biologists the dunes are referred to just as “substrate”, in Edaphology the dunes are considered as soils, therefore the soils of the studied area are classified as Arenosols (IUSS Working Group WRB, 2015).

In these Arenosols, the source of organic matter is transported by the wind together with other particles, it is not the result of in situ edaphic processes, for this reason, it is not related to the rest of the edaphic properties, instead, it is directly related to the height of the topoform, and inversely associated with the vegetation cover.

Magnetite content is evident in regional dune soils. Therefore, magnetite should be considered an important variable in forthcoming dune soils research.

Multivariate analysis (PCA and Cluster analysis) and the linear correlations showed that pH, fine sand, and magnetite influence the soil fertility of the embryo dunes of the Jalisco Coast, but not organic matter or moisture retention. These three edaphic properties are related to the regional geological source material, thus, is the most influential soil-forming factor in the incipient soils of the region.

From an edaphic perspective, it was possible to distinguish the distribution pattern of the exclusive species of embryo dunes concerning the ubiquitous species of coastal dunes, the former being less dependent on the edaphic conditions, and the latter being the most dependent on environmental conditions, especially on edaphic conditions. We assume that there is a plant succession between them, thus, the exclusive species of embryo dunes could be considered as pioneers. In addition to this, the PCA and the cluster analysis show that among the exclusive species of embryo dunes, some species prefer conditions of low fertility, while others prefer a higher degree of fertility.

Even in the context of its incipient edaphic and floristic development, regional embryo dune soils glimpse an interaction between plants, relief, and soil, which is reflected in the results of the two methods used for its understanding.

The development of this study made it possible to document the first report supported with research data on the relationship between environmental properties and plant species in the coastal dunes of Jalisco, México.

Supplemental Information

Supplemental Information 1 Ubiquitous plant species. (a) Ipomoea pes-capreae, (b) Jouvea pilosa.

Photo Credit: Miguel Ángel Macías-Rodríguez

Click here for additional data file.

Supplemental Information 2 Representative sites of the study area and their corresponding classification.

(a) Las Playitas, (b) El Coco, (c) La Limonera, (d) La Soledad, (e) North Chamela, (f) South El Tecuán. Photo Credit: Héctor Gerardo Frías-Ureña.

Click here for additional data file.

Supplemental Information 3 Dendrogram of environmental variables using average linkage (between groups).

Re-scaled distance cluster combine.

Click here for additional data file.

Supplemental Information 4 Dendrogram for the presence of plant species using Ward links.

Re-scaled distance cluster combine.

Click here for additional data file.

Supplemental Information 5 Raw data.

Click here for additional data file.

The authors express their gratitude to Roberto Esparza and Rafael Hernández, who collaborated in the fieldwork. We also thank Isaías Pedroza and Benjamín Hinojos for their participation in the lab determinations.

Additional Information and Declarations

Competing Interests

Author Contributions

Data Availability

The authors declare that they have no competing interests.

Héctor Gerardo Frías-Ureña conceived and designed the experiments, performed the experiments, analyzed the data, prepared figures and/or tables, authored or reviewed drafts of the paper, and approved the final draft.

José Ariel Ruiz-Corral analyzed the data, authored or reviewed drafts of the paper, and approved the final draft.

Miguel Ángel Macías-Rodríguez performed the experiments, authored or reviewed drafts of the paper, and approved the final draft.

Noé Durán analyzed the data, prepared figures and/or tables, and approved the final draft.

Diego Gonzalez conceived and designed the experiments, authored or reviewed drafts of the paper, and approved the final draft.

Fabio De Albuquerque analyzed the data, authored or reviewed drafts of the paper, and approved the final draft.

José Pablo Torres Morán analyzed the data, prepared figures and/or tables, and approved the final draft.

The following information was supplied regarding data availability:

The raw measurements are available in the Supplemental File.

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
