# Peer review of "Relationship between the distribution of vegetation and the environment in the coastal embryo dunes of Jalisco, México"

_PeerJ, doi:10.7717/peerj.13015_

## Round 0.1 · original submission · Major Revisions

Dear Dr. Ruiz-Corral,

Thank you for your submission to PeerJ.

It is my opinion as the Academic Editor for your article - Edaphic characterization of the coastal dunes of Jalisco, México, and its relationship with vegetation distribution - that it requires a number of Major Revisions.

Reviewers' comments on your work have now been received. The manuscript has been assessed by two reviewers. Reviews indicated that you need to use multivariate statistical methods like CCA, DCA, RDA or PCCA to show the real relationship. Moreover, "Discussion and conclusion" should also be improved to discuss in detail the relationship between vegetation and soil, vegetation and terrain, landform, and soil and terrain in the study area. I agree with this evaluation and I would, therefore, request for the manuscript to be revised accordingly.

My suggested changes and reviewer comments are shown below and on your article 'Overview' screen. In addition, one of the reviewers has attached an annotated manuscript to this review.

Please address these changes and resubmit. Although not a hard deadline please try to submit your revision within the next 35 days.

With kind regards,

Chenxi Li
Academic Editor, PeerJ

Reviewer 1 ·

Basic reporting

Edaphic characterization of the coastal dunes of Jalisco, México, and its relationship with vegetation distribution (#63863)
Introduction
The introduction section has several paragraphs that some of them should be merged (e.g., lns: 101-102 is a sentence!!!). Also, all citations used in the introduction is a bite old. Also, after reading the introduction, it is impossible to understand the gap and consequently the innovation of the paper.
Overall, introduction needs to improve by adding the background related to the problem statement of the study by using updated citations.

Experimental design

Materials and methods:
The sampling site (Lines 147-154) names should be added to the map of sampling sites.
Please clarify the date of sampling.
In the Laboratory section why only the sand fraction has been measured? As we know the silt or clay fractions can have strong correlation with chemical properties like OM.
Based on lines 205-212 the title of the manuscript should be changed to Environmental instead of edaphic!!!!
Statistical and spatial analysis: I think when we decided to investigate the relationship between environmental properties and vegetation types, we need to use multivariate statistical methods like CCA, DCA, or PCCA to show the real relationship. These methods combine both variables and it would be easy to determine the controlling parameters.

Validity of the findings

Results:
In the first section of results there are a set of correlation coefficient values and associated p-values, maybe it is better that bring them in a Table! And remove several paragraphs! The main point here is that all coefficients are weak correlations (less than 0.6; although the p-value is significant). When we review these results, we can see that sometimes 60% of data cannot explained by these coefficients: so, in discussion it should be explained, and the reasons must be found.
Table 1: the communalities is correct NOT Commonalities!!!
You should bring all PCs loading values as well as engine values, explained variance,…have a look on similar papers. Merge Table 1 &2.
Clustering the environmental shows that cluster 2 only has one site that you mentioned you collected 1- 2 samples in each site. So first you have to mention the number of samples in each cluster and although with 2 samples the statistical analysis should be meaningless.
Table 4: Group!!!!
Fig. 5&8: what is the meaning of the letters?

Additional comments

Discussion and conclusion: several paragraphs that I could not find the rational relation.
Overall, I think this manuscript needs a lot of works to be improved and suitable for publication.
Good luck

Reviewer 2 ·

Basic reporting

no comment

Experimental design

no comment

Validity of the findings

no comment

Additional comments

See attached PDF

Annotated reviews are not available for download in order to protect the identity of reviewers who chose to remain anonymous.

---

## Round 0.2 · accepted · Accept

Based on the previous round of review, the authors have addressed the concerns raised by the editor and reviewers.